# Impacts of COVID-19 Prevention and Control Measures on Asthma-Related Hospital and Outpatient Visits in Yichang, China

**DOI:** 10.3390/ijerph192013572

**Published:** 2022-10-20

**Authors:** Jinyi Wang, Dapeng Yin, Guoxing Li, Tianqi Wang, Yanli Zhang, Hui Gan, Jinfang Sun

**Affiliations:** 1Office of Epidemiology, Chinese Center for Disease Control and Prevention, Beijing 102206, China; 2Hainan Center for Disease Control and Prevention, Haikou 570110, China; 3Department of Occupational and Environmental Health Sciences, School of Public Health, Peking University, Beijing 100191, China; 4Institute for Scientific Information, Yichang Center for Disease Control and Prevention, Yichang 443000, China; 5The First Affiliated Hospital of Guangzhou Medical University, Guangzhou 510120, China

**Keywords:** COVID-19 lockdown, regular epidemic prevention and control, asthma, interruption time series, hospital and outpatient visits

## Abstract

This article investigates the impact of COVID-19 lockdown and regular epidemic prevention and control after lifting lockdown on asthma-related hospital and outpatient visits in Yichang. Data on the general outpatient department (GOPD), emergency department (ED) and intensive care unit (ICU) visits for asthma from 15 November 2019 to 21 May 2020 and the corresponding from 2018 to 2019 were collected from eight tertiary hospitals in municipal districts. The controlled interrupted time series (CITS) analysis was used to investigate the level and long-term trend changes of weekly asthma visits during lockdown and regular epidemic prevention and control, and stratified by type of visits and age. A total of 9347 asthma-related hospital and outpatient visits were analyzed. The CITS showed that after the implementation of lockdown, the weekly visits of asthma patients immediately decreased by 127.32 (*p* = 0.002), and the level of GOPD and ED/ICU visits immediately decreased significantly. After implementation of regular prevention, the level and trend of overall weekly visits changed insignificantly compared with the lockdown period. The weekly visits of GOPD adults immediately increased by 51.46 (*p* < 0.001), and the trend of ED/ICU adults decreased by 5.06 (*p* = 0.003) visits per week compared with lockdown period. The COVID-19 lockdown in Yichang was related to the decrease in hospital and outpatient visits for asthma. After the implementation of subsequent regular prevention and control measure, only the GOPD visits of adults increased compared with lockdown period.

## 1. Introduction

Asthma is a common chronic respiratory disease among children and adults, affecting more than 300 million people worldwide and causing nearly 350,000 deaths each year [1,2]. There are more than 45 million asthma patients in China. Recurrent attacks and continuous treatment of asthma have brought a huge burden to the health care system [3]. Respiratory viruses, inhaled allergens and air pollution represent significant environmental risk factors for asthma formation and acute attack [4,5,6]. Although this disease cannot be completely cured, the symptoms can be effectively controlled by maintaining the routine treatment in medical institutions, taking medicine regularly and reducing contact with its triggers. In addition, common comorbidities of asthma, such as rhinitis, sinusitis, respiratory tract infection, obesity and obstructive sleep apnea, can directly lead to hospital visits, complicate the diagnosis of asthma and increase the risk of misdiagnosis. For example, both obesity and asthma can cause dyspnea, and one study showed that 36% of asthma patients with obesity diagnosed by doctors may have a misdiagnosis of asthma [7].

Epidemic prevention and control have a significant impact on medical demand for most diseases [8], but the impact on asthma was rarely reported in China. A national retrospective survey reported that the prevalence of overweight and obesity has significantly increased in overall youths during COVID-19 lockdown in China [9]. Another study from Hubei (China) reported that adults who experienced social isolation during COVID-19 had significantly higher levels of generalized anxiety disorder symptoms [10]. A survey from 336 cities in China found that the all-cause, cardiovascular disease, respiratory disease and chronic obstructive pulmonary disease mortalities induced by NO_2_ decreased during the lockdown period [11]. In response to COVID-19, Yichang, Hubei Province activated first-level public health emergency response on 24 January 2020. Implementing mask wearing and strict social distancing measures, such as prohibiting indoor gatherings or events, suspending water and land transportation in municipal districts and counties and cities under its jurisdiction. Patients who needed face-to-face diagnosis or hospitalization were required to make an appointment in advance and inform the community before being admitted to designated hospitals for treatment. Since 16 March, the lockdown has been basically lifted, and the regular epidemic prevention and control phase has begun. The disinfection of key places was carried out comprehensively, the control of focus groups, places and traffic was strengthened, and residents were guided to take personal precautions such as wearing masks, practicing hygiene and maintaining social distancing. Hospitals had no other restrictions in the medical management system except demanding that fever patients visit fever clinics and requiring inpatients and accompanying staff to provide negative nucleic acid results. A series of prevention measures may affect patients’ medical treatment by changing the accessibility of medical services and the exposure risk of asthma environmental inducement. This study collected the information of asthma-related general outpatient department (GOPD), emergency department (ED) and intensive care unit (ICU) visits in eight tertiary hospitals in Yichang municipal districts during the epidemic prevention period and the same period of 2018–2019. In order to exclude the influence of comorbidities, only simple asthma patients were included. Analyses were also stratified by type of visits and age. The controlled interrupted time series (CITS) analysis was used to quantitatively evaluate the impact of lockdown and regular prevention and control after lifting lockdown on the level and long-term trend of weekly asthma visits.

Our current research shed light on several important but underexplored topics. Firstly, the study revealed the influence of two different degrees of COVID-19 prevention and control measures on asthma-related hospital and outpatient visits. In addition, analyses were also stratified by type of medical visits and age, in order to assess the impact of these measures on asthma-related medical visits of children and adults.

## 2. Materials and Methods

### 2.1. Data Sources and Setting

Yichang Healthcare Big Data Platform is managed by Yichang Center for Disease Control and Prevention (Yichang CDC), which contains medical and environmental indicators, birth and death registrations, and other health indicators. Yichang CDC is responsible for data collection, collation and quality control [12]. The big data platform has been widely used for epidemiological investigation and analysis [13,14]. Medical information includes encrypted ID number, gender, date of birth, ICD code and name of disease, comorbidity diagnosis, diagnosis time and department.

The frequency distribution data without individual information were collected from Yichang Healthcare Big Data Platform. Cases with diagnosis coded as J45 (asthma) and J46 (status asthmaticus) in the 10th revision of the International Statistical Classification of Diseases (ICD-10) were included. The data covered the period from 16 November 2018 to 21 May 2020, and included the GOPD, ED and ICU of all tertiary hospitals except the Special Care Hospital (Mental Health Center) in Yichang municipal districts (including 8 hospitals in total). Diagnostic records with missing gender and date of birth (0.04%), repeated entry (6.23%), and other comorbidities (24.92%) were excluded.

### 2.2. Research Methods

We conducted a controlled interrupted time series (CITS) analysis using data on weekly visits for asthma, including GOPD and ED/ICU visits. Yichang activated first-level public health emergency response on 24 January 2020 (Friday), so Friday was used as the beginning of each week to divide the weeks. The intervention group was from 15 November 2019 to 21 May 2020 (27 weeks), in which the lockdown intervention started on 24 January, and the regular prevention and control after lifting lockdown started on 13 March. Therefore, the research period was divided into pre-lockdown stage (10 weeks), lockdown stage (7 weeks) and regular prevention and control stage after lifting lockdown (10 weeks). The corresponding 27 weeks from 2018 to 2019 were selected as the control group, that is, from 16 November 2018 to 23 May 2019, which were also divided into 3 stages (see Figure 1 for details). In the CITS, we measured the impact of two public health interventions, respectively, accounting for comparative changes in the weekly visits in the intervention group relative to the control group. The control group can further control the time-varying confounding factors that are not included in the baseline trend (i.e., seasonal changes, meteorological events), thus isolating the net effect of intervention [15,16]. The CITS analysis was performed using a segmented regression model to analyze the immediate and long-term effects of these two interventions on weekly visits for simple asthma, which shown as the changes in level and trend (slope) before and after the intervention point. The statistical model was as follows:*Y_t_* = *β_0_* + *β_1_* × *time* + *β_2_* × *int_1_* + *β_3_* × *posttime* + *β_4_* × *group* + *β_5_* × *group* × *time* + *β_6_* × *group* × *int_1_* + β_7_ × *group* × *posttime* + *β_8_* × *int_2_* + *β_9_* × *Secondtime* + *β_10_* × *group* × *int_2_* + *β_11_* × *group* × *Secondtime* + *ε_t_*,

*Y_t_* was the number of weekly asthma visits; *time* was a time counting variable (0~26); *int_1_* was the intervention indicator variable of lockdown, and *int_2_* was the indicator of regular prevention and control after lifting lockdown (preintervention periods 0, otherwise 1); *Posttime* was the time counting variable after lockdown (0 before and 0~16 after intervention); *Secondtime* was the time counting variable after regular prevention and control after lifting lockdown (0 before and 0~9 after intervention); *Group* variable (the control group 0, otherwise 1); *ε_t_* was the residual.

*β_0_*~*β_3_* and *β_8_~β_9_* represented the control group, in which *β_0_* and *β_1_* were the intercept and slope of the outcome variable at the pre-lockdown stage; *β_2_* and *β_8_* were the changes in the level of outcome immediately following implementation of each intervention, respectively; *β_3_* and *β_9_* were the changes of slope before and after each intervention, respectively; *β_4_* was the difference in intercept between the two groups; *β_5_* was the difference in preintervention slope between the two groups; *β_6_* and *β_10_* were the differences between intervention and control groups in the level changes immediately (immediate changes) following introduction of each intervention, respectively; *β_7_* and *β_11_* were the differences in slope changes (trend changes) following each intervention between the two groups. The standard error was adjusted by Newey–West test to control autocorrelation and heteroscedasticity, and the normality of residual was tested by Shapiro–Wilk test.

Considering that patients in the ED and ICU were more seriously ill than those in the GOPD. In addition, children and adults differed in asthma treatment, severity and symptom relief, and their hospital visit may be differently affected by epidemic-related measures. Therefore, stratified the analysis by type of visits (GOPD, ED/ICU) and age (children < 18 years old, adults ≥ 18).

For basic descriptive statistics at the beginning, quantitative data of normal distribution were expressed as the mean values, and the significance of differences was evaluated by Student’s *t*-test. The statistical analysis was carried out with SAS software package (version 9.4, 100 SAS Campus Drive Cary, NC 27513), in adherence to a pre-defined significance level of 0.05 (two-sided).

## 3. Results

### 3.1. Baseline Characteristics

We evaluated a total of 9347 asthma-related hospital and outpatient visits during the study period: 4447 (47.58%) visits were men, 2557 (27.36%) visits were children under 18 years old, and 7517 (80.42%) visits in GOPD, 1830 (19.58%) in ED/ICU.

The total number of asthma visits in the intervention group was 3514, while that of the control group was 5833. At the pre-lockdown stage, the weekly visits of the two groups were basically parallel (Figure 1). During the lockdown period, the weekly visits in the intervention group decreased sharply and then remained at a low level (mean 50.71 visits/week), while that of the control revealed a trend of decrease first and then increase (mean 204.86 visits/week). At the regular prevention and control stage, the weekly visits in the intervention group increased and then remained stable (mean 118.60 visits/week), with no peak in early April as in the control group (mean 223.00 visits/week; Qingming Festival was in week 21 of Figure 1).

In the GOPD, during the lockdown period, the average weekly visits of both children and adults in the intervention group were significantly lower than pre-lockdown stage, while only that of children in the control group was significantly lower. At the regular prevention stage, both children and adults in two groups showed an increase, but only the intervention group had statistical significance.

In the ED/ICU, the average weekly visits of both children and adults in the control did not change significantly during lockdown. In the intervention group, the average weekly visits of both children and adults significantly declined during lockdown period. At the regular prevention stage, the average weekly visits of children in the intervention group decreased insignificantly, while that of adults increased significantly. Results are shown in Table 1.

### 3.2. Controlled Interrupted Time-Series Analysis

On the whole, there were no statistically significant differences in the baseline level and trend of weekly visit volumes during the pre-lockdown between two groups, and no significant changes in the level and trend after the two intervention points in the control group. In this study, the control group was used to control for time-varying confounders and isolate the net effects of interventions. The results showed that the number of weekly visits due to asthma immediately decreased by 127.32 (*p* = 0.002) after the implementation of lockdown, but the trend did not change significantly (*p* = 0.456). After the implementation of regular prevention and control, the weekly visit volumes immediately increased by 43.09, but with no statistical significance. There was no significant change in the trend (*p* = 0.490) during this stage compared with the lockdown period.

For GOPD visits of asthma in children, the baseline level and trend between two groups during the pre-lockdown were consistent, and there were no significant changes in level and trend after the two interventions in the control group. The weekly GOPD visits of children immediately decreased by 46.57 (*p* = 0.002) after the lockdown, while the changes of the level and trend after the regular prevention and control were nonsignificant. For GOPD visits of adults, the baseline level of weekly visits in the control group was 17.93 higher than that of the intervention group (*p* = 0.024), but both groups have the same trend during the pre-lockdown. The level and trend of the control group did not change significantly after the two intervention points. The results showed that the weekly GOPD visits of adults immediately decreased by 61.54 (*p* < 0.001) after the lockdown, and immediately increased by 51.46 (*p* < 0.001) after the start of regular prevention and control.

For ED/ICU visits of asthma in children, there were no significant changes in level and trend of weekly visit volumes after the implementation of lockdown or regular prevention and control. For ED/ICU visits of adults, the baseline level of weekly visits in the intervention group was 8.11 higher than the control (*p* = 0.003), and trend of the intervention group during pre-lockdown was significantly different from that of the control group (*p* < 0.001). In the control group, the weekly visit volumes showed an upward trend during pre-lockdown stage, with an increase of 3.41 visits per week (*p* < 0.001). At the lockdown stage, the trend of visit volumes decreased by 8.16 visits per week (*p* < 0.001), and then increased by 5.66 visits per week at the regular prevention and control stage (*p* < 0.001). The results indicated that the weekly ED/ICU visits among adults with asthma immediately decreased by 19.50 (*p* = 0.011) after the lockdown, and the trend significantly changed compared with the pre-lockdown period, increased by 7.14 (*p* < 0.001) visits per week. The decrease in level after the regular prevention and control was nonsignificant (*p* = 0.577), but the trend significantly changed compared with the lockdown stage, decreased by 5.06 (*p* = 0.003) visits per week. Results are shown in Table 2 and Figure 2.

## 4. Discussion

CITS analysis showed that the weekly hospital and outpatient visits of overall asthma patients in Yichang decreased greatly during the lockdown period, and it did not increase significantly after the regular prevention and control. After stratification, during the regular prevention period, only the weekly visits of adults increased significantly compared with lockdown period. Previous studies have focused on the impact of the COVID-19 lockdown on health care utilization for asthma, and mostly focus on children [17,18,19,20,21,22]. For example, a cross-sectional study from Guangzhou (China) reported that the number of hospitalized children with severe asthma exacerbation decreased under the implementation of strict countermeasures for COVID-19 [17]. Two related studies in Philadelphia, USA, also focused only on childhood asthma [18,19]. Although a study in the UK covered people of all ages, it only analyzed the impact of lockdown on asthma-related health care utilization, without further stratification by adults and children [20]. This study analyzed both the COVID-19 lockdown and regular prevention and control. In addition, asthma in children and adults are explored, respectively.

We found that after the implementation of lockdown in Yichang, GOPD and ED/ICU visits for simple asthma all decreased immediately, and no significant change was found in ED/ICU children, possibly due to the limited sample size. First, during the city lockdown, some hospitals in Yichang set up online clinics to provide basic medical services. Therefore, mild patients may tend to reduce face-to-face outpatient visits during the lockdown due to travel restrictions and fear of contracting the infection. However, patients who need emergency treatment or hospitalization can go to the designated hospitals after informing the community. Moreover, under the extreme conditions of lockdown, patients may be intensely focused on disease control and thereby increase medication compliance. Although asthma cannot be cured, adherence to regular medication can avoid repeated attacks. For example, in Wales, asthma-related oral and inhaled corticosteroid prescriptions in the week preceding the lockdown increased by 121% and 133%, respectively, compared with previous years [21]. To some extent, strict restrictions on social distance and public transportation during the lockdown have improved the air quality, and reduced the spread of respiratory viruses and the risk of asthma. A study on asthmatic children in Kobe showed a significant positive correlation between the ED visits volumes and SO_2_ levels, and the SO_2_ levels has dropped significantly after Japan declared a state of COVID-19 emergency [22]. During the lockdown in the USA, the concentration of traffic pollutant NO_2_ decreased by 46% and C_6_H_6_ decreased by 53% [23]. After the implementation of prevention and control measures such as Philadelphia Home Order, the positive rate of rhinovirus detection declined significantly [18].

Surprisingly, in control group, we observed a sharp decrease in the hospital visits for asthma among children and adults during early period of the corresponding stage of lockdown (1 February to 7 February 2019), which may be affected by the decline in visits during the Spring Festival. The average weekly visits of children from the control group decreased throughout the corresponding stage of lockdown, which may be affected by the school winter vacation. Because of the dense population in schools, it is easy to spread respiratory viruses. As a result, school closures during the winter vacation somewhat limit the spread of respiratory viruses, thus reducing the risk of asthma in children [24].

After the implementation of regular prevention and control, the weekly GOPD visits of asthma in adults immediately increased, but there was no significant change in children, and the weekly GOPD visits for both children and adults were lower than those during the previous year. After lifting lockdown, hospitals in Yichang had no other restrictions on medical visits except demanding fever patients to visit fever clinics and requiring inpatients and accompanying staff to provide negative nucleic acid results. The increase in adults suggests that the relaxation of COVID-19 prevention and control measures has led to a partial recovery of asthma-related routine medical services. However, there was no significant change in children, probably because childhood asthma is commonly associated with allergies. It is reported that the incidence of allergic asthma is highest in childhood and steadily decreases with advancing age. Therefore, children may be more susceptible to asthma than adults due to their sensitivity to allergens, but asthmatic children have a better chance of remission [25,26]. In this study, regular prevention and control was implemented from early March, just in the spring when the pollen allergens were prevalent. Wearing masks and other prevention and control measures reduced the exposure to asthma triggers such as allergens, which may be related to the decrease in GOPD visits for childhood asthma.

After implementation of regular prevention approaches, the trend of weekly ED/ICU visits for asthma in adults significantly declined compared with the lockdown stage, and neither children nor adults in ED/ICU experienced the peak of visits during Qingming Festival vacation (4 April to 6 April 2020, in the 21st week of the study) as in previous years. Because Qingming Festival vacation is the period of pollen transmission, spring outings increase the exposure to allergenic pollen, and crowd gathering also promotes the spread of respiratory viruses, so it is more likely to induce severe asthma, leading to the spike in hospital visits in previous years. During the epidemic prevention and control period, wearing masks can effectively isolate pollen allergens, which may lead to the disappearance of the peak of acute asthma visits during the vacation. In addition, a Finnish study showed that the number of ED visits caused by stroke and myocardial infarction remained stable before and after the epidemic prevention and control, while respiratory diseases decreased by 21.3%, indicating that patients who really need emergency treatment still visit medical institutions during this period [27]. The acute asthma attack in this study differs from acute diseases such as stroke and acute myocardial infarction, because patients can improve lung function and effectively control asthma attacks through self-management strategies, thus reducing the need for emergency medical services [28]. At the same time, ICU admission is based on the severity of illness, and ICU admission due to asthma has been identified as a risk factor for adverse asthma outcomes in the International Asthma Guidelines, which increases the risk of readmission for asthma [29]. Therefore, our findings suggest that there may be improvements in the control of severe asthma during regular prevention and control after lifting lockdown. Therefore, our findings suggest that regular prevention and control measures, such as mask-wearing and social distancing, may be related to the reduction in hospital visits for the acute exacerbation of asthma.

This study applied a CITS analysis, ITS has been considered as the strongest quasi-experimental design to causal inference and evaluate population-level public health interventions. ITS involves a pre-post comparison within the same population, and controlling for the counterfactual baseline trend. That is, assuming that the intervention did not happen, the baseline trend observed before the intervention were extrapolated to expected ongoing trend, and compared with the actual situation after the implementation of the intervention to assess the impact of intervention. The outcome variables are normally collected at equal time intervals, and the fluctuation trend before the intervention (baseline trend) is controlled. Statistical models are used to evaluate the intervention effect, including level and long-term trend changes. A change in the level refers to an immediate increase or decrease in the outcome after the intervention and represents an abrupt intervention effect. A change in trend refers to an increase or decrease in the slope of the segment after intervention compared to the pre-intervention segment. The ITS is suitable for our study because we used routine medical data, which are greatly appropriate in ITS studies. In addition, the time of intervention was clearly defined to distinguish the effects of different components. Furthermore, the outcome variables of our study were expected to change relatively quickly after the intervention, and ITS works best with short-term outcomes. Considering that time series data are often affected by potential time-varying confounders that are not included in the baseline trend, for example, other events that occurred around the time of the intervention that may affect the outcome (i.e., seasonal trends, holiday effects, meteorological events) [16]. We used CITS analysis, which added a control group that was not exposed to intervention. CITS can further control the influence of potential time-varying confounding factors and isolate the net effect of intervention, so it provides stronger evidence to support a causal relationship between the intervention and outcome. Therefore, compared with traditional ecological studies, the conclusions of CITS analysis are more reliable [30,31]. Moreover, considering the nonspecific symptoms of asthma, related comorbidities can increase the risk of misdiagnosis of asthma and directly affect patients’ hospital visits. Therefore, this study excluded comorbidities, and only studied the impact of epidemic prevention and control on hospital visits for patients with simple asthma. 

We analyzed the short-term and long-term effects of regular epidemic prevention and control on the hospital visits of asthma, and relevant domestic investigations are still lacking. The study has the following limitations: first, considering the different diagnostic levels of medical and health institutions at various levels, and the Yichang Healthcare Big Data Platform does not completely cover primary and secondary medical institutions, only tertiary hospitals in municipal districts were included for analysis. However, some asthma patients may have diverted to community health service centers during the epidemic prevention and control period, resulting in biased results. Additionally, it was difficult to find a contemporaneous control group that was not affected by epidemic prevention and control. Therefore, asthma patients in the same period of adjacent year were selected as a control, so that the social environment, seasonal influences and demographic characteristics of each group were as similar as possible, and the two groups were comparable.

## 5. Conclusions

This study provided important baseline data on the changes of asthma-related hospital and outpatient visits during public health interventions in COVID-19. Our findings suggest that the COVID-19 lockdown in Yichang was related to the decrease in hospital and outpatient visits for asthma. After the implementation of subsequent regular prevention and control measure, only the GOPD visits of adults increased compared with lockdown period. Although our research cannot clearly prove the reasons for the persistently low number of visits, we believe that a series of health-related measures, such as wearing masks and social distance (both included in lockdown and regular prevention and control policies), may have a positive impact on asthma control. Further work is needed to investigate the specific factors responsible for such decrease. 

## Figures and Tables

**Figure 1 ijerph-19-13572-f001:**
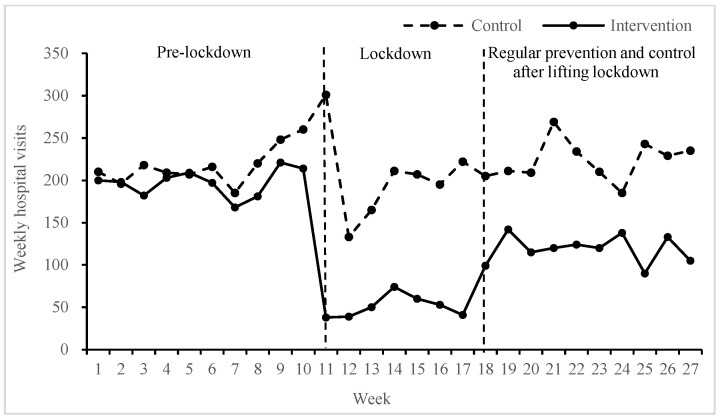
Weekly visits of asthma patients and distribution of epidemic prevention and control measures in Yichang.

**Figure 2 ijerph-19-13572-f002:**
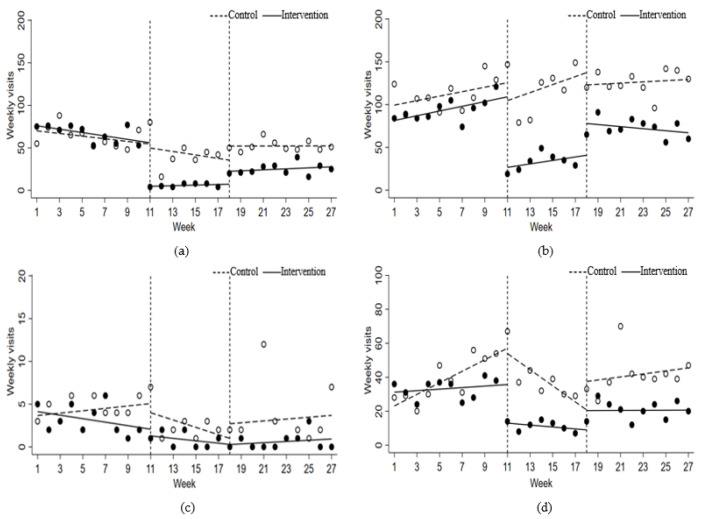
Changes in weekly visits of patients with asthma. (**a**) GOPD visits among children with asthma; (**b**) GOPD visits among adults with asthma; (**c**) ED/ICU visits among children with asthma; (**d**) ED/ICU among adults with asthma.

**Table 1 ijerph-19-13572-t001:** Hospital and outpatient visits of asthma patients at each stage [average weekly visits].

Object	Control					Intervention				
	Pre- Lockdown	Lockdown	Regular Prevention and Control	^a^*p*-Value	^b^*p*-Value	Pre- Lockdown	Lockdown	Regular Prevention & Control	^a^*p*-Value	^b^*p*-Value
GOPD										
children	63.00	43.71	52.20	0.023	0.299	67.00	5.86	25.00	< 0.0001	< 0.0001
adults	111.20	118.71	126.20	0.515	0.536	93.90	32.71	72.50	< 0.0001	< 0.0001
ED/ICU										
children	4.30	2.71	3.20	0.078	0.754	3.20	0.86	0.60	0.004	0.587
adults	38.40	39.71	41.40	0.839	0.780	33.20	11.29	20.50	< 0.0001	0.001

^a^ Comparison of average weekly visits between Pre-lockdown and Lockdown stages by Student’s *t*-test. ^b^ Comparison of average weekly visits between Lockdown and Regular prevention and control stages by Student’s *t*-test.

**Table 2 ijerph-19-13572-t002:** CITS analysis of asthma-related hospital and outpatient visits per week before and after COVID-19 preventions [parameter estimates (95%CI)].

Variable	Total Patient, *n* = 9347	GOPD Children, *n* = 2419	GOPD Adults, *n* = 5098	ED/ICU Children, *n* = 136	ED/ICU Adults, *n* = 1692
constant term	196.15 (180.51~211.78) *	69.98 (55.38~84.58) *	99.42 (85.05~113.79) *	3.84 (1.98~5.70) *	23.07 (17.38~28.77) *
*time*	4.61 (0.25~8.97) *	−1.55 (−4.09~0.98)	2.62 (−0.40~5.64)	−0.03 (−0.33~0.27)	3.41 (2.40~4.41) *
*int_1_*	−29.80 (−98.05~38.44)	−4.64 (−30.32~21.02)	−20.92 (−47.79~5.95)	−1.18 (−3.29~0.94)	−3.17 (−15.65~9.31)
*posttime*	−7.15 (−27.54~13.25)	−0.48 (−7.45~6.48)	2.06 (−5.62~9.74)	0.32 (−0.20~0.83)	−8.16 (−10.98~−5.34) *
*group*	−3.34 (−21.90~15.20)	6.02 (−9.08~21.12)	−17.93 (−33.40~−2.45) *	0.29 (−2.24~2.82)	8.11 (0.81~15.40)*
*group × time*	−3.61 (−9.05~1.82)	−0.45 (−3.24~2.34)	0.14 (−3.21~3.49)	−0.18 (−0.61~0.25)	−2.96 (−4.39~−1.53)*
*group × int_1_*	−127.32 (−203.81~−50.83) *	−46.57 (−74.14~−19.00) *	−61.54 (−91.93~−31.15) *	0.40 (−2.73~3.52)	−19.50 (−34.20~−4.80) *
*group × posttime*	7.83 (−13.15~28.80)	2.84 (−4.25~9.93)	−2.78 (−10.95~5.38)	−0.25 (−0.90~0.40)	7.14 (4.01~10.26) *
*int_2_*	20.38 (−42.05~82.80)	16.52 (−3.71~36.75)	−14.45 (−34.88~5.98)	−2.44 (−4.92~0.04)	16.60 (−1.01~34.20)
*Secondtime*	4.29 (−13.49~22.07)	2.06 (−3.84~7.96)	−3.96 (−10.76~2.84)	−0.001 (−0.48~0.47)	5.659 (2.66~8.66) *
*group × int_2_*	43.09 (−22.66~108.83)	−1.42 (−22.05~19.20)	51.46 (27.63~75.29) *	2.43 (−0.40~5.25)	−5.23 (−24.01~13.55)
*group × Secondtime*	−6.48 (−25.25~12.28)	−1.84 (−7.85~4.18)	0.73 (−6.84~8.30)	0.22 (−0.37~0.80)	−5.06 (−8.34~−1.77) *
*R^2^*	0.867	0.838	0.825	0.368	0.749

* *p* < 0.05; constant term: intercept of control group; time: slope (trend) of control group before lockdown; *int_1_* & *posttime*: changes of level and slope in the control group immediately after lockdown; *group* and *group* × *time*: difference of intercept (baseline level) and slope between two groups before lockdown; *int_2_* & *Secondtime*: changes of level and slope in control group immediately after regular prevention; *group* × *int_1_* and *group* × *posttime*: changes in the level and slope of weekly asthma visits after lockdown, reflecting the instantaneous and long-term effects of lockdown after controlling the time-varying confounding factors with control group; *group* × *int_2_* and *group* × *Secondtime*: changes in the level and slope following introduction of regular prevention and control after lifting lockdown.

## Data Availability

The data are not publicly available due to institutional and privacy restrictions.

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
