# Peer review of "Impacts of COVID-19 Prevention and Control Measures on Asthma-Related Hospital and Outpatient Visits in Yichang, China"

_ijerph, 2022, doi:10.3390/ijerph192013572_

Round 1

Reviewer 1 Report

Jinyi Wang et al. present an epidemiology study assessing the impact of COVID-19 prevention and control measures on asthma-related medical visits in Yichang, China. The data source includes general outpatient department (GOPD), emergency department (ED) and intensive care unit (ICU) visits for asthma in patients – adults and children without comorbidities. 

The authors used controlled interrupted time series 19 (CITS) analysis to provide objectivity.

The manuscript is well organized with high scientific soundness and quality. 

The paper could be accepted without further revision. 

I have a minor comment considering the article title - Asthma-related Hospital Visits usually indicate hospital admission, including emergency department (ED) and intensive care unit (ICU) visits. However, the study includes data from outpatient visits. These visits are usually associated with milder symptoms without the need for hospitalization. Therefore, it is more correct to be pointed to Asthma-related Hospital and Outpatient Visits instead of Asthma-related Hospital Visits.

Reviewer 2 Report

General Comment

The author investigated the relationship between prevention and control measures for the COVID-19 outbreak and asthma-related hospital visits in Yichang municipal districts (Hubei province). This study revealed that lockdown measures decreased asthma-related hospital visits, on the other hand, they increased at the regular prevention and control stage. The author used controlled interrupted time series (CITS) analysis to reveal this phenomenon. In addition, the author analyzed the data by stratifying the type of hospital visits and age category. 

The result of this study is important in aspects of epidemiological impacts on asthma during the COVID-19 outbreak. However, some description is complicated, and the author should change these descriptions into comprehensive ones. 

Specific Comments

Major

Abstract

Line 23-27: The author describes the results of the study as relative values in the abstract. However, the descriptions in the result were mainly absolute values. I guess that these differences in the description make it obscure. The author should unify the description. 

2. Materials and Methods

2.1 Data Sources

Line 73-83: In this chapter, the author described that the data was collected Yichang health big data platform, including 8 tertiary hospitals. I guess that some hospitals limited medical care in the ED or ICU department because they had to deal with many patients with COVID-19. The author should explain the change in the medical management system in these facilities.  

2. Results

2.1 Baseline characteristics

Line 139-141: It is difficult to understand the description of the results after stratification in GOPD and ED/ICU visits. The author should clarify what they compare. They should clearly state whether the description of “two groups” refers to adults and children, control and intervention. It is also necessary to state whether the change in the control group during the pre-lockdown and the lockdown periods can be a statistically significant reduction. 

Line 141-142: The author described that both GOPD children and adults in two groups showed a reduction at the regular prevention stage. But Table 1 shows that weekly GOPD visits increased for both children and adults in both the control and intervention groups. I guess that this description is incorrect. 

Line 143-144: The author described that the average weekly visits of children in the control decreased during the lockdown period. The author should add the result of the statistical analysis. The same goes for adults’ results.

5. Conclusion

Line 314-316: The author describes that the COVID-19 lockdown and regular epidemic prevention and control after lifting lockdown in Yichang were related to the decrease of hospital visits for asthma. This description is inconsistent with the ones in the abstract and results. The author should unify the description of the results in this study. 

Minor

The serial number of the contents (“2. Results” → “3. Results”) is incorrect. The author should the serial numbers. 

Reviewer 3 Report

The study investigated the impact of COVID 19 lockdown and reuglar epidemic prevention and control on asthma-related hospital visits in Yichang, China. More than 9,000 asthma-related hospital visits were analyzed. The controlled interrupted time series analysis shows that after lockdown in 20220, the weekly visits decreased significantly. After implementation of regular prevention approaches, the visits increased bY >50% in the general out-patient department by decreased by 5% in the emergency department and in ICU. The conclusion of the authors is that covid-19 lockdown and teh implementation of prevention policy results in a decrease in hospital visits for asthma patients. 

While the study is interesting, and appears to be properly conducted, the conclusions of the authors are not unexpected and actually rather obvious. This reviewer is failing to appreciate how the information provided in the study is enhancing our understanding in the field, and which useful health-related policy is originated from by the results. There is no indication in that sense in the manuscript. 

Round 2

Reviewer 2 Report

The author corrected the manuscript based on the reviewers' suggestions appropriately. I could not indicate any mistakes or corrections in the revised manuscript. 

Reviewer 3 Report

This is a revised version of a study addressing the impact of Covid-19 lockdown in Yichang on outpatient visits for asthma. The results of the study indicate that immediately after the implementation of the lockdown the weekly visits of outpatients with asthma decreased significanlty. Following the implementatino of preventive measures, the reduced trend of visits continued although a 50% increase was observed in visits by adults patients (but not in children) as compared to the lockdown levels. THis increase, however, remained below the level prior to the occurrence of COvid pandemic.